# Virulence genes, antimicrobial resistance profile, phylotyping and pathotyping of diarrheagenic *Escherichia coli* isolated from children in Southwest Mexico

**Gabriela Tapia-Pastrana**[1]*, **Metztli Rojas-Bautista**[2], **Pilar Hernández-Pérez**[2], **Olegario Santiago-Martínez**[2], **Lucía C. Gómez-Rodríguez**[2], **Víctor M. Terrazas-Luna**[1], **Jacobo Montes-Yedra**[3], **Alfonso A. Bautista-Avendaño**[3], **Eduardo S. García-López**[4], **Nidia Leon-Sicairos**[5,6], **Uriel A. Angulo-Zamudio**[5], **Adrian Canizalez-Roman**[5,7]

**1** Laboratorio de Investigación Biomédica, Hospital Regional de Alta Especialidad de Oaxaca, Servicios de Salud, IMSS-Bienestar, Oaxaca, Mexico, **2** Facultad de Ciencias Químicas de la Universidad Autónoma Benito Juárez de Oaxaca, Oaxaca de Juarez, Oaxaca, Mexico, **3** Departamento de Ciencias Básicas del Instituto Tecnológico del Valle de Oaxaca, Santa Cruz Xoxocotlán, Oaxaca, Mexico, **4** Laboratorio Estatal de Salud Publica Oaxaca, Reyes Mantecón, Oaxaca, Mexico, **5** School of Medicine, Autonomous University of Sinaloa, Culiacan, Sinaloa, Mexico, **6** Pediatric Hospital of Sinaloa, Culiacan, Sinaloa, Mexico, **7** The Women's Hospital, Secretariat of Health, Culiacan, Sinaloa, Mexico

* gtapia@hraeoaxaca.gob.mx

## Abstract

Diarrheagenic *E. coli* (DEC) strains are one of the most important etiology factors causing diarrhea in children worldwide, especially in developing countries. DEC strains have characteristic virulence factors; however, other supplemental virulence genes (SVG) may contribute to the development of diarrhea in children. Therefore, this study aimed to determine the prevalence of DEC in children with diarrhea in southwestern Mexico and to associate childhood symptoms, SVG, and pathotypes with diarrhea-causing DEC strains. DEC strains were isolated from 230 children with diarrhea aged 0–60 months from the state of Oaxaca, southwestern Mexico; clinical data were collected, and PCR was used to identify SVG and pathotypes. Antibiotic resistance profiling was performed on DEC strains. 63% of samples were DEC positive, single or combined infections (two (21%) or three strains (1.3%)) of aEPEC (51%), EAEC (10.2%), tEPEC (5.4%), DAEC (4.8%), ETEC (4.1%), EIEC (1.4%), or EHEC (0.7%) were found. Children aged $\leq$ 12 and 49–60 months and symptoms (e.g., fever and blood) were associated with DEC strains. SVG related to colonization (*nleB*-EHEC), cytotoxicity (*sat*-DAEC and *espC*-tEPEC), and proteolysis (*pic*-aEPEC) were associated with DECs strains. *E. coli* phylogroup A was the most frequent, and some pathotypes (aEPEC—A, DAEC–B), and SVG (*espC*–B2, and *sat*–D) were associated with the phylogroups. Over 79% of the DEC strains were resistant to antibiotics, and 40% were MDR and XDR, respectively. In conclusion aEPEC was the most prevalent pathotype in children with diarrhea in this region. SVG related to colonization, cytotoxicity, and proteolysis were associated with diarrhea-producing DEC strains, which may play an essential role in the development of diarrhea in children in southwestern Mexico.

**Data Availability Statement:** All relevant data are within the manuscript and its Supporting Information files.

**Funding:** This work was supported by grants #25501 and #25101 from the Secretaría de Salud through the Hospital Regional de Alta Especialidad de Oaxaca to G.T.P.

**Competing interests:** The authors have declared that no competing interests exist.

## Introduction

Diarrhea is a serious public health problem and a significant cause of child morbidity and mortality; More than 446,000 child deaths from diarrhea have been reported worldwide [1]. Developing countries are most affected by this disease, including those in Africa, Asia, and Latin America, with most cases associated with child wasting, unsafe sanitation, unsafe water sources, poor water conditions, and poor cooking practices, among others [1, 2]. Several etiological agents cause diarrhea; however, diarrheagenic *E. coli* (DEC) is one of the most common.

DEC are classified according to their production of virulence factors, some related to the adhesion process, cytotoxic capacity, or whether they contain a capsule [3]. The presence of specific virulence factor genes classifies DEC into six different pathotypes, and this information is shown in S1 Table.

In addition to the virulence factors that characterize each DEC pathotype, some supplementary virulence genes (SVG) have recently been associated with the ability of DEC to cause diarrhea and with the higher virulence phenotype of these strains [4]. SVG are defined as genes encoding proteins involved in (i) colonization (*ehaC*, *ehaD*, *cah*, and *nleB*, which are associated with adhesins or biofilm formation [5, 6]), (ii) cytotoxicity (*sat*, *espC*, *pet*, *hylA*, and *stx1*, these toxins are associated with damage to the cytoskeleton, tight junctions or ribosome of epithelial cells [7, 8]), (iii) proteolysis (*pic*, *eatA*, and *espP*, which encode proteins with the ability to degrade mucin, spectrin, pepsin or plasma proteins [9, 10]), (iv) cyclomodulins (*cnf*, *pks*, and *cif*, which encode proteins with the ability to cause DNA damage, modulate the cell cycle, and induce epithelial cell tumors [11]), and v) iron acquisition (*ybtS*, and *entB*, these genes have activities related to siderophores and iron metabolism [12]).

Another essential feature of DEC is the high antibiotic resistance of these bacteria. DEC isolated from children with diarrhea has increased antibiotic resistance over the years; for example, in the early 2000s, European countries reported zero resistance to trimethoprim, but in 2016, this resistance increased to 28.8% [13, 14]. Antibiotic resistance in *E. coli* also varies across nations; in Organization for Economic Cooperation and Development (OECD) countries, *E. coli* resistance ranged from 0.3% to 37.7%, while in non-OECD countries, *E. coli* antibiotic resistance ranged from 5% to 80% [13, 14]. Antibiotic resistance is a significant public health problem that is increasing yearly, so more strategies must be developed to address this problem.

DEC strains are associated with cases of diarrhea, but the population most affected by these bacteria are children. Cases of DEC diarrhea in children have been reported worldwide in the United States of America, Mexico, South America, Europe, and Asia [15–19]. However, studies have shown that DEC may also require SVG to develop cases of diarrhea; a study conducted in the state of Yucatan showed that DEC with SVG (*aap*, *aatA*, *astA*, *pet*, and *cdt*) resulted in more cases of moderate or severe diarrhea in children [4]. In addition, Angulo-Zamudio et al., (2021) found that DEC were associated with diarrhea when the strains carried an SVG (*ehaC*, *kps*, *nleB*, and/or *espC*) in children in northwestern Mexico [20].

Furthermore, Oaxaca is one of the three poorest states in Mexico, and the cause of DEC diarrhea, DEC-associated signs, and symptoms, and DEC-associated SVG leading to diarrhea may differ from the rest of the Mexican states. Therefore, this comprehensive study evaluated the prevalence of DEC in Oaxacan children aged 0 to 60 months, symptoms, SVG, pathotypes, and their association with diarrhea-causing DEC strains. Finally, the antibiotic resistance profiles of the DEC strains were studied.

## Materials and methods

### Sample collection

From January 1, 2016, to January 31, 2017, a total of 230 stool samples were collected from children aged 0–5 years of both sexes with diarrhea from six health jurisdictions in the state of Oaxaca located in southwestern Mexico: Valles Centrales (J1), Istmo (J2), Tuxtepec (J3), Costa (J4), Mixteca (J5), and Sierra Sur (J6) (Fig 1). The state of Oaxaca is one of the most impoverished in Mexico, and the population of these six jurisdictions represents ~99% of the state's inhabitants [21]. Patients with acute diarrhea came to primary care units where they were evaluated by licensed physicians who confirmed that patients had the passage of three or more loose or liquid stools per day or more frequent passage than is usual. In addition, other symptoms were recorded from electronic medical records: fever, vomiting, dehydration, abdominal pain, and watery diarrhea with mucus and/or blood. Rectal swabs were collected by standard procedures for routine diagnostics at the primary care units and transferred to the Cary-Blair transport medium. Swabs were stored at 4°C at the communal health station until transportation in cold boxes to the Public Health Laboratory of the State of Oaxaca and the Biomedical Research Laboratory of the Hospital Regional de Alta Especialidad de Oaxaca (HRAEO), usually within 2 hours. For the sample collection, the children who participated in the project, the parents signed an informed consent form explaining the sample collection procedures, and the

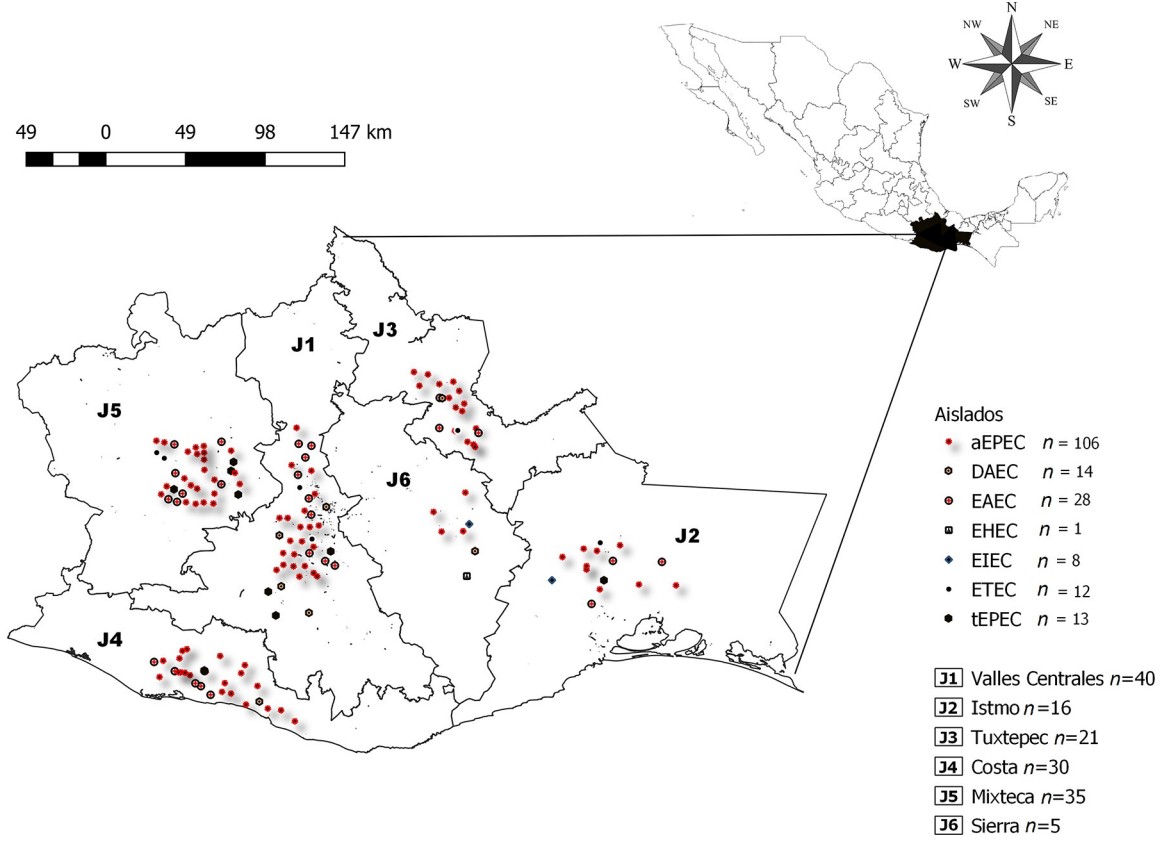

**Fig 1. Geographical locations of sampling sites and pathotypes identified by the municipality of Oaxaca state, southwest Mexico.** The open-source software QGIS (Quantum GIS Development Team) was used to pinpoint the geographic locations of the sampling sites and can be found at https://qgis.org/es/site/.

study was approved by the Ethics Committee of the HRAEO, which approved this project No. HRAEO-CIC-CEI 017/14.

## Isolation and identification of *E. coli*

Samples were streaked on MacConkey agar for 24 hours at 37˚C, and colonies with *E. coli*-related characteristics were subjected to the following conventional biochemical tests: catalase test (+), oxidase test (-), glucose fermentation with gas production. In addition, the biochemical test was performed using the API20E kit (Biomerieux, Marcy-l'Etoile, France) according to the manufacturer's instructions. In addition, a PCR assay was performed to identify the 16SrRNA gene of *E. coli* for molecular identification. The sequence of the primers and the size of the PCR products are shown in S2 Table. Finally, *E. coli* isolates were subcultured on Luria-Bertani (LB) agar and stored in LB broth plus 20% glycerol at -70˚C for further experiments.

## Bacterial strains

The reference strains used in this study were from our laboratory collection. EPEC E2348/69 (*eae+* and *bfp+*), EHEC O157:H7 EDL933 (*eae+*, *stx1+* and *stx2+*), EIEC EI-34 (*ipaH+* and *virF+*), EAEC O42 (*aafII+*), ETEC H10407 (*lt+*) and DAEC F1845 (*daaE+*) were used to identify DEC. For the identification of SVG, cyclomodulins and siderophores, strains used in previous studies of our group were used as controls [20]. In addition, the most representative amplicons were sequenced for control purposes.

## Preparation of template DNA

The boiling method was used to extract genomic DNA. Three bacterial colonies from each isolate were vortexed in a 1.5 mL tube containing nuclease-free water. After boiling for 5 minutes, the mixture was immediately placed in a freezer for 2 minutes. The suspension was then centrifuged (12,000 × g) for 5 min. From the supernatant (containing the extracted DNA), 150 μL of the sample was transferred to a new 0.6 mL tube. The DNA was stored at -20˚C for further experiments.

## Molecular identification of DEC strains

DEC strains among *E. coli* isolates were identified using a multiplex and singleplex PCR reaction scheme based on previous work [20, 22]: Multiplex PCR reaction #1 contained primers to amplify the intimin gene (*eae*) and the structural subunit of the bundle-forming pilus gene (*bfp*), and detected both typical (tEPEC, *eae+* and *bfp+*) and atypical (aEPEC, *eae+* and *bfp-*) EPEC strains. Multiplex PCR reaction #2 detected EHEC strains and included primers to amplify the *eae* gene and the Shiga toxin genes *stx1* and *stx2*. EIEC strains were detected with reaction #3, which included primers to amplify the invasion genes *virF* and *ipaH*. EAEC strains were detected by amplification of the *aafII* gene using singleplex PCR reaction number 4. ETEC strains were detected using reaction number 5, which contained primers to amplify the *lt* gene. Finally, singleplex PCR reaction number 6 had primers amplifying the *daaE* gene to detect DAEC strains. The sequences of the primers and the sizes of the PCR products with these targets are shown in S2 Table. For all PCR reactions, we mixed MyTaq (BIOLINE, Tennessee, USA), 0.5 μM direct and reverse primers, and 2 μL genomic DNA and molecular biology grade water, and the reactions were performed on a Mastercycler flexlid thermal cycler (Eppendorf, Hamburg, Germany). PCR products were visualized with ethidium bromide, and the results were analyzed on a PhotoDoc-It imaging system with a Benchtop 2 UV transilluminator (UVP). All DEC-negative specimens were excluded from the study.

## Molecular identification of the phylogenetic groups

Phylogenetic groups A, B1, B2, and D were identified as described by Clermont et al. We performed singleplex PCR reactions to identify the *chuA*, *yjaA*, and *tspE4.C2* genes (Clermont, Bonacorsi, and Bingen, 2000). Primers sequences and PCR product sizes for these targets are listed in S2 Table.

## Detection of SVG, cyclomodulins, and siderophores by PCR

The SVG targets were divided into three groups based on the putative role of the proteins encoded by each gene in virulence. The SVG related to colonization were *ehaC*, *ehaD*, *cah*, and *nleB*. Those related to cytotoxicity: *pet*, *espC*, *sat*, *hlyA* and *stx1*. Those related to proteolysis: *pic*, *eatA*, and *espP*. cyclomodulins were also identified: Pks, CNF, and CIF (*clbB*, *cnf*, and *cif* were used as target genes). Finally, the siderophore-associated genes *ybtS* and *entB* were identified. Primers sequences and PCR product sizes are listed in S2 Table.

## Cytotoxicity assay

Cytotoxicity was evaluated in HEp-2 cells (laryngeal carcinoma cells) using isolates of the different DEC pathotypes with randomly selected SVG. HEp-2 cells were cultured in DMEM medium (Corning) supplemented with 10% fetal bovine serum and antibiotics [penicillin 100 U/mL, streptomycin 100 μg/mL] (Gibco). Cells were incubated in a 5% CO2 atmosphere at 37°C until 90% confluence was reached. HEp-2 cells were then detached with 0.25% trypsin, EDTA, and 1X sodium bicarbonate (Corning), resuspended in supplemented DMEM, and seeded in a 96-well box (Costar 3590, CORNING) at a density of 30,000 cells/mL and grown to monolayer confluence using the incubation conditions described above. Infection was performed at a multiplicity of infection (MOI) of 10 for 12 h in a 5% CO2 atmosphere at 37°C. *E. coli* HB101 was used as a negative control for cytotoxicity, while buffer lysis was used as a positive control. Cytotoxicity was quantified by LDH release using the CytoTox 96® kit (PROMEGA) according to the manufacturer's instructions.

## Antimicrobial susceptibility testing

Antimicrobial susceptibility testing of pathogenic isolates was performed using the Kirby-Bauer disk diffusion method according to the guidelines developed by the Clinical Laboratory Standard Institute, following the methodology of previous work [20]. Suspensions of *E. coli* isolates were prepared in LB at 0.5 turbidity using the McFarland standard. These cultures were then seeded on Mueller-Hinton agar plates, and antibiotic discs (BD BBL, Franklin Lakes, NJ, USA) were aseptically applied to the inoculated agar. The antibiotics tested were ampicillin, amoxicillin/clavulanic acid, ampicillin/sulbactam, cefazolin, cefotaxime, ceftriaxone, ceftazidime, aztreonam, imipenem, kanamycin, streptomycin, tetracycline, ciprofloxacin, nalidixic acid, trimethoprim, and chloramphenicol. Plates were incubated at 37°C for 18 h. The diameters (in millimeters) of the clear growth inhibition zones around the antimicrobial discs were measured using a precision digital caliper (Absolute, Mitutoyo, Japan). *E. coli* (Migula) Castellani and Chalmers (ATCC® 25922™) was used as a control. The diameters of the inhibition halos were then measured using a ruler. Antibiotic susceptibility was interpreted according to CLSI guidelines; bacteria were classified as resistant, intermediate, or sensitive [23]. Isolates resistant to ≥ 3 different categories of antibiotics were classified as Multidrug-resistant (MDR), and those resistant to ≥ 6 different categories of antibiotics were classified as Extremely drug-resistant (XDR) [24]. Antibiotics were selected based on their use in treating

human infections caused by Gram-negative bacteria and represent different classes of antimicrobial agents available for treating these infections in Mexico [25].

## Statistical analysis

Associations between nominal variables were analyzed using Fisher's exact test and/or chi-squared. Differences in the level of cytotoxicity of DEC strains were determined by one-way analysis of variance, with a p-value ≤0.05 considered statistically significant; IBM® SPSS® Statistics version 20 (New York, USA) was used to perform the statistical analyses, while Sig-maPlot version 12 (CA, USA) was used to construct the graphs.

## Results

### Prevalence of DEC pathotypes in children with diarrhea

Of the 230 samples collected from children with diarrhea, we analyzed 147; 83 diarrhea samples were excluded, three of which were positive for *Salmonella* or *Shigella flexnerii*, and the rest were not DEC. Sixty-three percent (147/230) of the samples were positive for at least one *E. coli* pathotype, and their distribution in the six health jurisdictions of the state of Oaxaca is shown in Fig 1.

Children were divided into five age groups (≤12, 13–24, 25–36, 37–48, and 49–60 months), 46.3% (68/147) were ≤12 months, 18.4% (27/147) 13–24 months, 4.1% (6/147) 25–36 months, 2% (3/147) 37–48 months and 29.3% (43/147) 49–60 months (Table 1). Seven pathotypes were individually identified in children with diarrhea; the most common pathotype found in the samples was aEPEC 51% (75/147), followed by EAEC 10.2% (15/147), in smaller proportions tEPEC (5.4%, 8/147), DAEC (4.8%, 7/147), ETEC (4.1%, 6/147), EIEC (1.4%, 2/147) and EHEC (0.7%, 1/147) (Table 1). In addition, co-infections of two or three DEC were found in the samples, seven co-infections of two pathotypes, and the most common was EAEC+aEPEC 7.5% (11/147), followed by DAEC+aEPEC 4.1% (6/147), the other five combinations (ETEC+aEPEC, EIEC+aEPEC, aEPEC+tEPEC, tEPEC+EIEC and EAEC+DAEC) were found in lower proportion, Table 1. As for the co-infection of three pathotypes, ETEC+aEPEC+EAEC and EIEC+tEPEC+aEPEC were found in one sample each (Table 1). In addition, the isolate-by-isolate PCR profile is shown in S3 Table.

Regarding the presence of pathotypes in age groups, aEPEC was the most common pathotype in all age groups (Table 1). However, three pathotypes were associated with age groups, EAEC (16.2%, 11/68) and DAEC (8.8%, 6/68) were associated with children ≤12 months (*p*: 0.026 and *p*: 0.031, respectively), whereas the presence of tEPEC (11.6%, 5/43) was associated with children aged 49–60 months (*p*: 0.033) (Table 1). No associations with any age group were found between other single pathotypes or co-infection (two or three pathotypes).

### Association between DEC and clinical characteristics of children with diarrhea

Several symptoms were recorded in infected children, including fever, vomiting, abdominal pain, and others, and associations with DEC causing diarrhea were sought; individual DEC infection was analyzed as DEC co-infection, individual symptoms, and diarrhea + 1–4 additional symptoms (Table 2). In individual DEC infection, vomiting was the most frequent symptom in children with aEPEC, EAEC and DAEC (52%, 39/75, 66.7%, 10/15 and 71.4%, 5/7, respectively), dehydration, watery diarrhea, bloody diarrhea in tEPEC (50%, 4/8), watery diarrhea in children infected with ETEC (83.3%, 5/6), EIEC and EHEC showed the same prevalence in all symptoms (Table 2). As for diarrhea + 1–4 symptoms, the distribution of symptoms was similar among all DEC pathotypes. Although several symptoms were identified in

**Table 1. Diarrheagenic *Escherichia coli* (DEC) isolated from children with diarrhea by age group.**

| *E. coli* pathotypes | Age group (months) | | | | | | |
|---|---|---|---|---|---|---|---|
| | Total children | ≤ 12 (%) | 13–24 (%) | 25–36 (%) | 37–48 (%) | 49–60 (%) | *p*-value |
| | n = 147 (%) | n = 68 (46.3) | n = 27 (18.4) | n = 6 (4.1) | n = 3 (2) | n = 43 (29.3) | |
| **Single pathotype** | | | | | | | |
| aEPEC | 75 (51.01) | 31 (45.6) | 17 (63) | 3 (50) | 1 (33.3) | 23 (53.5) | 0.169 |
| EAEC | 15 (10.2) | 11 (16.2) * | 2 (7.4) | 0 (0.0) | 1 (33.3) | 1 (2.3) | 0.026 |
| tEPEC | 8 (5.4) | 3 (4.4) | 0 (0.0) | 0 (0.0) | 0 (0.0) | 5 (11.6) * | 0.033 |
| DAEC | 7 (4.8) | 6 (8.8) * | 1 (3.7) | 0 (0.0) | 0 (0.0) | 0 (0.0) | 0.031 |
| ETEC | 6 (4.1) | 3 (4.4) | 1 (3.7) | 0 (0.0) | 0 (0.0) | 2 (4.7) | 0.822 |
| EIEC | 2 (1.4) | 1 (1.5) | 0 (0.0) | 0 (0.0) | 0 (0.0) | 1 (2.3) | 0.516 |
| EHEC | 1 (0.7) | 0 (0.0) | 1 (3.7) | 0 (0.0) | 0 (0.0) | 0 (0.0) | 0 (0.0) |
| **DEC co-infections** | | | | | | | |
| EAEC+aEPEC | 11 (7.5) | 3 (4.4) | 1 (3.7) | 1 (16.7) | 1 (33.3) | 5 (11.6) | 0.085 |
| DAEC+aEPEC | 6 (4.1) | 2 (2.9) | 2 (7.4) | 1 (16.7) | 0 (0.0) | 1 (2.3) | 0.111 |
| ETEC+aEPEC | 5 (3.4) | 3 (4.4) | 1 (3.7) | 0 (0.0) | 0 (0.0) | 1 (2.3) | 0.53 |
| EIEC+aEPEC | 4 (2.7) | 3 (4.4) | 0 (0.0) | 0 (0.0) | 0 (0.0) | 1 (2.3) | 0.242 |
| aEPEC+tEPEC | 3 (2.0) | 1 (1.5) | 0 (0.0) | 1 (16.7) | 0 (0.0) | 1 (2.3) | 0.875 |
| tEPEC+EIEC | 1 (0.7) | 1 (1.5) | 0 (0.0) | 0 (0.0) | 0 (0.0) | 0 (0.0) | 0.279 |
| EAEC+DAEC | 1 (0.7) | 0 (0.0) | 0 (0.0) | 0 (0.0) | 0 (0.0) | 1 (2.3) | 0.118 |
| ETEC+aEPEC+EAEC | 1 (0.7) | 0 (0.0) | 0 (0.0) | 0 (0.0) | 0 (0.0) | 1 (2.3) | 0.118 |
| EIEC+tEPEC+aEPEC | 1 (0.7) | 0 (0.0) | 1 (3.7) | 0 (0.0) | 0 (0.0) | 0 (0.0) | 0 (0.0) |
| **Number Pathotype** | | | | | | | |
| 1 | 114 (77.5) | 55(89.9) | 22(81.5) | 3(50) | 2(66.7) | 32(74.4) | 0.369 |
| 2 | 31 (21.0) | 13(19.1) | 4(14.8) | 3(50) | 1(33.9) | 10(23.3) | 0.076 |
| 3 | 2 (1.36) | 0 (0.0) | 1(3.7) | 0 (0.0) | 0 (0.0) | 1(2.3) | 0.244 |

OR: odds ratio, CI: confidence index

*: statistical significance. Chi-square was used to get statistical significance.

different DEC, only bloody diarrhea was associated with EIEC (*p*: 0.001), and no association was found between the rest of the individual DEC and symptoms (Table 2).

Regarding DEC co-infection, watery diarrhea was the most common symptom in EAEC + aEPEC and ETEC+ aEPEC infected children (63.3%, 7/11 and 80%, 4/5, respectively), fever in DAEC+ aEPEC (66.7%, 4/6), abdominal pain in EIEC+ aEPEC (75%, 3/4), mucoid diarrhea in aEPEC+ tEPEC (66. 7%, 2/3), all other combinations of two or three DEC showed the same prevalence in all symptoms; of all individual symptom totals with DEC co-infection, only fever was associated with DAEC+ aEPEC (*p*: 0.032) compared to all other pathotype combinations (Table 2). As for diarrhea + 1–4 symptoms, there was some association with DEC combinations, diarrhea + 1 symptom was associated with ETEC + aEPEC (40%, 2/5, p: 0.002), diarrhea + 2 symptoms with aEPEC+ tEPEC (33. 3%, 1/3, p: 0.018), diarrhea + 3 symptoms with EAEC + aEPEC (27.3%, 3/11, p: 0.0001) and diarrhea + 4 symptoms with DAEC+ aEPEC (50%, 3/6, *p*: 0.0001) compared to all other pathotype combinations (Table 2).

## Supplementary Virulence Genes (SVG) in DEC isolated from children with diarrhea

After analyzing the 147 children with diarrhea and possible associations with DEC and some characteristics of the children (age and symptoms), we identified SVG in the DEC strains

**Table 2. Association between diarrheagenic *Escherichia coli* (DEC) and clinical symptoms of children with diarrhea.**

| Clinical Symptoms | Individual DEC infection | | | | | | | | DEC co-infections | | | | | | | | | |
| --- | --- | --- | --- | --- | --- | --- | --- | --- | --- | --- | --- | --- | --- | --- | --- | --- | --- | --- |
| | aEPEC n=75 (%) | EAEC n=15 (%) | tEPEC n=8 (%) | ETEC n=6 (%) | DAEC n=7 (%) | EIEC n=2 (%) | EHEC n=1 (%) | p value | EAEC +aEPEC n=11 (%) | DAEC +aEPEC n=6 (%) | ETEC +aEPEC n=5 (%) | EIEC +aEPEC n=4 (%) | EIEC +tEPEC n=1 (%) | EAEC +DAEC n=1 (%) | aEPEC +tEPEC n=3 (%) | EIEC +aEPEC +tEPEC n=1 (%) | ETEC +EAEC +aEPEC n=1 (%) | p value |
| Fever | 27 (36.0) | 5 (33.0) | 0 (0.0) | 3 (50.0) | 3 (42.9) | 0 (0.0) | 1 (100) | 0.402 | 3(27.3) | 4(66.7)* | 0 (0.0) | 2(50) | 1(100) | 0 (0.0) | 0 (0.0) | 0 (0.0) | 0 (0.0) | 0.032* |
| Vomiting | 39 (52.0) | 10 (66.7) | 3 (37.5) | 3 (50.0) | 5 (71.4) | 1 (50.0) | 1 (100) | 0.35 | 7(63.6) | 5(83.3) | 2(40) | 2(50) | 1(100) | 1(100) | 1(33.3) | 0 (0.0) | 0 (0.0) | 0.158 |
| Abdominal pain | 23 (30.7) | 4(26.7) | 1(12.5) | 4 (66.7) | 3 (42.9) | 0 (0.0) | 1 (100) | 0.057 | 4(36.4) | 4(66.7) | 2(40) | 3(75) | 1(100) | 0 (0.0) | 1(33.3) | 1(100) | 0 (0.0) | 0.257 |
| Dehydration | 38 (50.7) | 9(60) | 4(50.0) | 3 (50.0) | 4 (57.1) | 1 (50.0) | 1 (100) | 0.539 | 6(54.5) | 5(83.3) | 3(60) | 2(50) | 1(100) | 0 (0.0) | 1(33.3) | 0 (0.0) | 0 (0.0) | 0.117 |
| Watery diarrhea | 37 (49.3) | 9(60) | 4(50.0) | 5 (83.3) | 4 (57.1) | 0 (0.0) | 0 (0.0) | 0.111 | 7(63.6) | 5(83.3) | 4(80) | 1(25) | 0 (0.0) | 1(100) | 1(33.3) | 0 (0.0) | 0 (0.0) | 0.158 |
| Mucus diarrhea | 34 (45.3) | 6(40) | 4(50.0) | 1 (16.7) | 3 (42.9) | 1 (50.0) | 1 (100) | 0.716 | 3(27.3) | 1(16.7) | 1(20) | 2(50) | 0 (0.0) | 0 (0.0) | 2(66.7) | 1(100) | 1(100) | 0.199 |
| Bloody diarrhea | 4(5.3) | 0 (0.0) | 0 (0.0) | 0 (0.0) | 0 (0.0) | 1 (50.0) | 1 (100) | 0.001* | 1(9.1) | 0 (0.0) | 0 (0.0) | 1(25) | 1(100) | 0 (0.0) | 0 (0.0) | 0 (0.0) | 0 (0.0) | 0.237 |
| Only diarrhea | 21(28) | 2(13.3) | 3(37.5) | 0 (0.0) | 1 (14.3) | 1(50) | 0 (0.0) | 0.171 | 3(27.3) | 0 (0.0) | 1(20) | 1(25) | 0 (0.0) | 0 (0.0) | 1(33.3) | 0 (0.0) | 1(100) | 0.018 |
| *Diarrhea + any symptom* | | | | | | | | | | | | | | | | | | |
| 1 | 12(16) | 4(26.7) | 3(37.5) | 2 (33.3) | 2 (28.6) | 0 (0.0) | 0 (0.0) | 0.08 | 1(9.1) | 1(16.7) | 2(40)* | 0 (0.0) | 0 (0.0) | 1(100) | 1(33.3) | 1(100) | 0 (0.0) | 0.002 |
| 2 | 18(24) | 4(26.7) | 1(12.5) | 2 (33.3) | 0 (0.0) | 1(50) | 0 (0.0) | 0.305 | 3(27.3) | 1(16.7) | 1(20) | 1(25) | 0 (0.0) | 0 (0.0) | 1(33.3)* | 0 (0.0) | 0 (0.0) | 0.018 |
| 3 | 17(22) | 4(26.7) | 1(12.5) | 1 (16.7) | 3 (42.9) | 0 (0.0) | 0 (0.0) | 0.073 | 3(27.3)* | 1(16.7) | 1(20) | 1(25) | 0 (0.0) | 0 (0.0) | 0 (0.0) | 0 (0.0) | 0 (0.0) | 0.0001 |
| 4 | 7(9) | 1(6.7) | 0 (0.0) | 1 (16.7) | 1 (14.3) | 0 (0.0) | 1 (100) | 0.382 | 1(9.1) | 3(50.0) | 0 (0.0) | 1(25) | 1(100) | 0 (0.0) | 0 (0.0) | 0 (0.0) | 0 (0.0) | 0 |

OR: odds ratio, CI: confidence index

*: statistical significance. Chi-square was used to get statistical significance.

isolated from the children. Of the 147 diarrheal samples, we isolated 182 DEC, of which aEPEC was the most prevalent DEC with 58.2% (106/182), followed by EAEC 15.3% (28/182), in smaller proportions DAEC (7.1%, 13/182), tEPEC (6.5%, 12/182), ETEC (6.5%, 12/182), EIEC (4.3%, 8/182), and EHEC (0.5%, 1/182) (Table 3).

Seventeen SVG related to colonization (n = 4), cytotoxicity (n = 5), proteolysis (n = 3), cyclomodulins (n = 3), and iron acquisition (n = 2) were identified in the 182 DEC isolated from children with diarrhea. The distribution of SVG carried by the strains was broad in the DEC; in those related to colonization, of the total DEC, *ehaC* was most prevalent (91.2%, 166/182), followed by *ehaD* (61.5%, 112/182), *cah* (5.4%, 10/182) and *nleB* (4.3%, 8/182); by the number of genes, most strains carried 1 to 2 genes (more than 80%). Regarding pathotype, only the presence of *nleB* was associated with EHEC (100%, 1/1, *p*: 0.001); by the number of genes, the presence of 0 and 4 genes related to SVG colonization were associated with EIEC (25%, 2/8, *p*: 0.044) and aEPEC (1.8%, 2/106, *p*: 0.001), respectively (Table 4).

Regarding cytotoxicity-related genes, *sat* was the most common (25.2%, 46/182), followed by *espC* (22.5%, 41/182), *pet* (7.1%, 13/182), *hlyA*, and *stx1* (0.5%, 1/182), respectively; by number of genes, 51% (93/182) were negative for all genes, while the remaining DEC were positive for 1–2 SVG. *sat*, and e*spC* were associated with DAEC (57%, 8/14, *p*: 0.004) and tEPEC (76.9%, 10/13, *p*: 0.001), respectively, except for EIEC, which was negative for all cytotoxicity-related SVG (*p*: 0.012), the distribution by the number of these SVG was similar in all pathotypes (Table 3). For proteolysis-related SVG, *pic* was the most common (11.5%, 21/182), followed by *eatA* (6%, 11/182) and *espP* (0.5%, 1/182); 87.9% of DEC were negative for these SVG, and the remaining strains were positive for 1–2 genes. EAEC was associated with *pic* (25%, 7/28, *p*: 0.01) and one gene (28.5%, 8/28, *p*: 0.02); the distribution of the remaining SVG in DEC was similar (Table 3).

On the other hand, the presence of cyclomodulins in DEC was limited, *cnf* being the most abundant (9.8%, 18/182), then *pks* (4.3%, 8/182) and *cif* (2.1%, 4/182); 75.2% (137/182) of the strains were negative for any gene, while 16. 3% were positive for 1–2 genes. In this SVG group, no association was found with *E. coli* pathotypes by gene or gene number (Table 3). Similar to the cyclomodulins, in the genes related to iron acquisition, there was no association between these genes (*ybtS*, and *entB*) or the number of these genes with DEC pathotypes (Table 3).

## Cytotoxicity of DEC strains with SVG in human cells

To determine the cytotoxic potential of DEC isolated from children with SVG diarrhea, we tested these strains with HEp-2 cells, and cytotoxic activity was assessed by LDH quantification using a commercial kit. We randomly selected DEC strains positive for some SVG of each identified pathotype, EAEC (n = 3), EHEC (n = 1), EPEC (n = 6), ETEC (n = 2), and DAEC (n = 2), as a positive damage control, lysis buffer, and E were used. Overall and in their pathotype group, EAEC1 (*ehaC+*, *ehaD+*, *pet+*, and *ytbs+*) and EAEC2 (*ehaC+*, *pet+*, and *ytbs+*) were the most cytotoxic (over 100% and 70%, respectively). Although EHEC harbored seven SVG, it presented a low cytotoxic activity (<40%); it was similar to that of *E. coli* HB101; as for EPEC, EPECa3 strain (*ehaC+*, *sat+*, and *ytbs+*) presented the highest cytotoxic activity (about 60%) compared to the rest of EPEC (Fig 2). As for ETEC, ETEC1 (*ehaC+*, *ehaD+*, *eatA+*, and *entB+*) presented higher levels of LHD (50%) compared to ETEC2; similar behavior in DAEC, strain DAEC1 (*ehaC+*, *ehaD+*, *sat+*, and *ytbs+*) was the most cytotoxic compared to DAEC2 (Fig 2).

## Distribution of *E. coli* phylogroups and their association with supplemental virulence genes (SVG)

Different *E. coli* phylogroups (A, B1, B2, and D) were identified by PCR in DEC isolated from children with diarrhea; the most prevalent was A (n = 75), followed by B2 (n = 48), D (n = 38)

**Table 3. Distribution of SVG in DEC isolated from children with diarrhea.**

| SVG implied on: | Total DEC | aEPEC (%) | EAEC (%) | DAEC (%) | tEPEC (%) | ETEC (%) | EIEC (%) | EHEC (%) | *p* |
|---|---|---|---|---|---|---|---|---|---|
| | n = 182 (%) | n = 106 (58.2) | n = 28 (15.3) | n = 14 (7.9) | n = 13 (7.1) | n = 12 (6.5) | n = 8 (4.3) | n = 1 (0.5) | value |
| **Colonization** | | | | | | | | | |
| *ehaC* | 166 (91.2) | 98 (92.5) | 27 (96.4) | 13 (92.9) | 10 (76.9) | 11 (91.7) | 6 (75) | 1 (100) | 0.289 |
| *ehaD* | 112 (61.5) | 69 (65.1) | 14 (50) | 10 (71.4) | 6 (46.2) | 9 (75) | 4 (50) | 0 (0.0) | 0.321 |
| *cah* | 10 (5.4) | 4 (3.8) | 0 (0.0) | 0 (0) | 1 (7.7) | 0 (0.0) | 0 (0.0) | 0 (0.0) | 0.257 |
| *nleB* | 8 (4.3) | 7 (6.6) | 1 (3.6) | 0 (0) | 0 (0.0) | 0 (0.0) | 0 (0.0) | 1 (100)* | 0.001 |
| by number of genes | | | | | | | | | |
| 0 | 13 (7.1) | 5 (4.7) | 1 (3.5) | 1 (7.1) | 3 (23.0) | 1 (8.3) | 2 (25.0)* | 0 (0.0) | 0.044 |
| 1 | 59 (32.4) | 34 (32.0) | 13 (46.4) | 3 (21.4) | 4 (30.7) | 2 (16.6) | 2 (25.0) | 1 (100) | 0.08 |
| 2 | 100 (54.9) | 59 (55.6) | 13 (46.4) | 10 (71.42) | 5 (38.4) | 9 (75.0) | 4 (50.0) | 0 (0.0) | 0.14 |
| 3 | 8 (4.9) | 6 (5.6) | 1 (3.5) | 0 (0.0) | 1 (7.6) | 0 (0.0) | 0 (0.0) | 0 (0.0) | 0.54 |
| 4 | 2 (1.0) | 2 (1.8)* | 0 (0.0) | 0 (0.0) | 0 (0.0) | 0 (0.0) | 0 (0.0) | 0 (0.0) | 0.001 |
| **Cytotoxicity** | | | | | | | | | |
| *sat* | 46 (25.2) | 26 (24.5) | 6 (21.4) | 8 (57.0)* | 2 (15.4) | 4 (33.3) | 0 (0.0) | 0 (0.0) | 0.004 |
| *espC* | 41 (22.5) | 31 (29.2) | 0 (0.0) | 0 (0.0) | 10 (76.9)* | 0 (0.0) | 0 (0.0) | 0 (0.0) | 0.001 |
| *pet* | 13 (7.1) | 0 (0.0) | 13 (46.4) | 0 (0.0) | 0 (0.0) | 0 (0.0) | 0 (0.0) | 0 (0.0) | - |
| *hlyA* | 1 (0.5) | 0 (0.0) | 0 (0.0) | 0 (0.0) | 0 (0.0) | 0 (0.0) | 0 (0.0) | 1 (100) | - |
| *stx 1* | 1 (0.5) | 0 (0.0) | 0 (0.0) | 0 (0.0) | 0 (0.0) | 0 (0.0) | 0 (0.0) | 1 (100) | - |
| by number of genes | | | | | | | | | |
| 0 | 93 (51.0) | 57 (53.7) | 11 (39.2) | 6 (46.8) | 3 (23.0) | 8 (66.6) | 8 (100)* | 0 (0.0) | 0.012 |
| 1 | 76 (41.7) | 41 (38.6) | 15 (53.5) | 8 (57.1) | 8 (61.5) | 4 (33.3) | 0 (0.0) | 0 (0.0) | 0.13 |
| 2 | 12 (6.5) | 8 (7.5) | 2 (7.1) | 0 (0.0) | 2 (15.3) | 0 (0.0) | 0 (0.0) | 1 (100) | 0.23 |
| 3 | 0 (0.0) | 0 (0.0) | 0 (0.0) | 0 (0.0) | 0 (0.0) | 0 (0.0) | 0 (0.0) | 0 (0.0) | - |
| 4 | 0 (0.0) | 0 (0.0) | 0 (0.0) | 0 (0.0) | 0 (0.0) | 0 (0.0) | 0 (0.0) | 0 (0.0) | - |
| 5 | 0 (0.0) | 0 (0.0) | 0 (0.0) | 0 (0.0) | 0 (0.0) | 0 (0.0) | 0 (0.0) | 0 (0.0) | - |
| **Proteolysis** | | | | | | | | | |
| *pic* | 21 (11.5) | 14 (13.2)* | 7 (25) | 0 (0.0) | 0 (0.0) | 0 (0.0) | 0 (0.0) | 0 (0.0) | 0.01 |
| *eatA* | 11 (6.0) | 8 (7.5) | 1 (3.6) | 0 (0.0) | 0 (0.0) | 2 (16.7) | 0 (0.0) | 0 (0.0) | 0.11 |
| *espP* | 1 (0.5) | 0 (0.0) | 0 (0.0) | 0 (0.0) | 0 (0.0) | 0 (0.0) | 0 (0.0) | 1 (100) | - |
| by the number of genes | | | | | | | | | |
| 0 | 160 (87.9) | 87 (82.0) | 20 (71.4) | 14 (100) | 13 (100) | 10 (83.3) | 8 (100) | 0 (0.0) | 0.08 |
| 1 | 27 (14.8) | 16 (15.0) | 8 (28.5) | 0 (0.0) | 0 (0.0) | 2 (16.6) | 0 (0.0) | 1 (100) | 0.02 |
| 2 | 3 (1.6) | 3 (2.8) | 0 (0.0) | 0 (0.0) | 0 (0.0) | 0 (0.0) | 0 (0.0) | 0 (0.0) | 0.13 |
| 3 | 0 (0.0) | 0 (0.0) | 0 (0.0) | 0 (0.0) | 0 (0.0) | 0 (0.0) | 0 (0.0) | 0 (0.0) | - |
| **Cyclomodulin** | | | | | | | | | |
| *cnf* | 18 (9.8) | 9 (8.5) | 5 (19.9) | 1 (7.1) | 2 (15.4) | 0 (0.0) | 1 (12.5) | 0 (0.0) | 0.12 |
| *pks* | 8 (4.3) | 4 (3.8) | 3 (10.7) | 0 (0.0) | 1 (7.7) | 0 (0.0) | 0 (0.0) | 0 (0.0) | 0.07 |
| *cif* | 4 (2.1) | 2 (1.9) | 0 (0.0) | 0 (0.0) | 1 (7.7) | 0 (0.0) | 0 (0.0) | 1 (100) | 0.16 |

(*Continued*)

**Table 3.** (Continued)

| SVG implied on: | Total DEC | aEPEC (%) | EAEC (%) | DAEC (%) | tEPEC (%) | ETEC (%) | EIEC (%) | EHEC (%) | *p* |
|---|---|---|---|---|---|---|---|---|---|
| | n = 182 (%) | n = 106 (58.2) | n = 28 (15.3) | n = 14 (7.9) | n = 13 (7.1) | n = 12 (6.5) | n = 8 (4.3) | n = 1 (0.5) | value |
| by the number of genes | | | | | | | | | |
| 0 | 137 (75.2) | 93 (87.7) | 22 (78.5) | 13 (92.8) | 10 (76.9) | 12 (100) | 7 (87.5) | 0 (0.0) | 0.15 |
| 1 | 20 (10.9) | 11 (10.3) | 4 (14.2) | 1 (7.14) | 2 (15.3) | 0 (0.0) | 1 (12.5) | 1 (100) | 0.59 |
| 2 | 10 (5.4) | 2 (1.8) | 2 (7.1) | 0 (0.0) | 1 (7.6) | 0 (0.0) | 0 (0.0) | 0 (0.0) | 0.25 |
| 3 | 0 (0.0) | 0 (0.0) | 0 (0.0) | 0 (0.0) | 0 (0.0) | 0 (0.0) | 0 (0.0) | 0 (0.0) | - |
| **Iron acquisition** | | | | | | | | | |
| *ybtS* | 102 (56.0) | 64 (60.4) | 15 (53.6) | 6 (42.9) | 8 (61.5) | 6 (50) | 3 (37.5) | 0 (0.0) | 0.67 |
| *entB* | 23 (12.6) | 12 (11.3) | 5(17.9) | 2 (14.3) | 2 (15.4) | 2 (16.7) | 0 (0.0) | 0 (0.0) | 0.36 |
| by number of genes | | | | | | | | | |
| 0 | 73 (40.1) | 40 (37.7) | 11 (39.2) | 7 (50) | 4 (30.7) | 5 (41.6) | 5 (62.7) | 1 (100) | 0.18 |
| 1 | 93 (51.0) | 56 (52.8) | 14 (50) | 6 (42.8) | 8 (61.5) | 6 (50) | 3 (37.5) | 0 (0.0) | 0.43 |
| 2 | 16 (8.7) | 10 (9.4) | 3 (10.7) | 1 (7.1) | 1 (7.6) | 1 (8.3) | 0 (0.0) | 0 (0.0) | 0.69 |

SVG: supplementary virulence genes, CI: confide index

*: statistical significance. Chi-square was used to get statistical significance.

and B1 (n = 21) (Table 4). Regarding pathotypes and phylogroup, aEPEC was the most prevalent in all phylogroups and was associated with phylogroup A (69.3%, 52/75, *p*: 0.011); similarly, DAEC was associated with phylogroup B2 (14.5%, 7/48, p: 0.036) compared to the rest of the phylogroups (Table 4). The remaining pathotypes showed a similar distribution in all phylogroups with no associations.

Regarding the relationship between *E. coli* phylogroups and SVG, we found some associations Table 2. In genes related to colonization, *ehaC*, and *ehaD* were present in more than 75% and 50% of the phylogroups, respectively, while *cah* was present in phylogroups B1 and B2, as well as *nleB*, which was found in A and B1, despite the distribution of this group of genes in the phylogroups of *E. coli*, no associations were found (Table 4).

Among cytotoxicity-related genes, *pet* was found only in phylogroup A, while espC was associated with B2 (76.9%, 10/13, p: 0.001), also sat was associated with phylogroup D (57%, 8/14, p: 0. 004), *hlyA* and *stx 1* were not found in any phylogroup (Table 4). *pic* was associated with phylogroup A (25%, 7/28, p: 0.01), *espP* was present in phylogroup B2, while *eatA* was present only in A and B1; this in genes related to proteolysis (Table 4). As for cyclomodulin-related genes, *cnf* was found in all phylogroups, *pks* in A, B1, and B2, while *cif* was only in B1 and B2; no associations were found between cyclomodulin and phylogroups (Table 4). Finally, both genes (*entB*, and *ybts*) were found in all *E. coli* phylogroups, but no associations were found (Table 4).

## Antimicrobial resistance of *E. coli* strains isolated from children with diarrhea

Antimicrobial resistance information for DEC strains (n = 182) isolated from children with diarrhea (n = 147) is shown in Table 5. Overall, more than 50% of the strains were resistant to

**Table 4. Distribution of phylogroup in *E. coli* isolated from children with diarrhea.**

| *E. coli* pathotypes | A n = 75 | B1 n = 21 | B2 n = 48 | D n = 38 | *p* |
|---|---|---|---|---|---|
| | n (%) | n (%) | n (%) | n (%) | value |
| aEPEC | 52(69.3)* | 12 (57.1) | 19 (39.5) | 23 (60.5) | 0.011 |
| DAEC | 2 (2.6) | 1 (4.7) | 7 (14.5)* | 4 (10.5) | 0.036 |
| tEPEC | 4 (5.3) | 2 (9.5) | 5 (10.4) | 2 (5.2) | 0.409 |
| ETEC | 4 (5.3) | 1 (4.7) | 5 (10.4) | 2 (5.2) | 0.180 |
| EIEC | 5 (6.6) | 1 (4.7) | 1 (2.0) | 1 (4.3) | 0.187 |
| EAEC | 8 (10.6) | 4 (19.0) | 10 (20.8) | 6 (15.7) | 0.222 |
| EHEC | 0 (0.0) | 0 (0.0) | 1 (2.0) | 0 (0.0) | 0 |
| **SVG implied on:** | **n = 28 (%)** | **n = 106 (%)** | **n = 13 (%)** | **n = 14 (%)** | ***p*-value** |
| **Colonization** | | | | | |
| *ehaC* | 27 (96.4) | 98 (92.5) | 10 (76.9) | 13 (92.9) | 0.289 |
| *ehaD* | 14 (50) | 69 (65.1) | 6 (46.2) | 10 (71.4) | 0.321 |
| *cah* | 0 (0.0) | 4 (3.8) | 1 (7.7) | 0 (0.0) | 0.257 |
| *nleB* | 1 (3.6) | 7 (6.6) | 0 (0.0) | 0 (0.0) | 0 |
| **Cytotoxicity** | | | | | |
| *pet* | 13 (46.4) | 0 (0.0) | 0 (0.0) | 0 (0.0) | 0 |
| *espC* | 0 (0.0) | 31 (29.2) | 10 (76.9)* | 0 (0.0) | 0.001 |
| *sat* | 6 (21.4) | 26 (24.5) | 2 (15.4) | 8 (57.0)* | 0.004 |
| *hlyA* | 0 (0.0) | 0 (0.0) | 0 (0.0) | 0 (0.0) | 0 |
| *stx 1* | 0 (0.0) | 0 (0.0) | 0 (0.0) | 0 (0.0) | 0 |
| **Proteolysis** | | | | | |
| *pic* | 7 (25) * | 14 (13.2) | 0 (0.0) | 0 (0.0) | 0.01 |
| *espP* | 0 (0.0) | 0 (0.0) | 1 (7.7) | 0 (0.0) | 0 |
| *eatA* | 1 (3.6) | 8 (7.5) | 0 (0.0) | 0 (0.0) | 0.11 |
| **Cyclomodulin** | | | | | |
| *pks* | 3 (10.7) | 4 (3.8) | 1 (7.7) | 0 (0.0) | 0.07 |
| *cnf* | 5 (19.9) | 9 (8.5) | 2 (15.4) | 1 (7.1) | 0.12 |
| *cif* | 0 (0.0) | 2 (1.9) | 1 (7.7) | 0 (0.0) | 0.16 |
| **Iron acquisition** | | | | | |
| *entB* | 5(17.9) | 12 (11.3) | 2 (15.4) | 2 (14.3) | 0.36 |
| *ybtS* | 15 (53.6) | 64 (60.4) | 8 (61.5) | 6 (42.9) | 0.67 |

SVG: supplementary virulence genes, CI: confide index

*: statistical significance. Chi-square was used to get statistical significance

cefazolin (89.5%), ampicillin (71.4%), streptomycin (64.2%), and trimethoprim (55.4%). By pathotype, EAEC was statistically significantly ($p$: ≤0.05) more resistant to cefotaxime (60.7%), ceftriaxone (28.5%), streptomycin (82.1%), while DAEC was resistant to amoxicillin-clavulanic acid, ampicillin-sulbactam (57. 1% each) and tetracycline (75.8%), ETEC to cefazolin (100%) and imipenem (25%), EIEC to cefazolin (100%) and chloramphenicol (25%); finally, EHEC to cefazolin (100%) (Table 5). By number of antibiotics, 73.9% of DEC were resistant to 1–7 antibiotics, while the rest were resistant to 8–14 antibiotics (Table 5). By category, 79.7% were resistant to any antibiotic, while 3.8% were sensitive to any antibiotic, 43.9% were MDR, and 41.2% were XDR; by pathotype, tEPECs had more antibiotic-sensitive bacteria (23%, $p$: 0.008), while ETECs had more MDR (75%, $p$: 0.026) and DAECs had higher prevalence of XDR (64.2%, $p$: 0.047) compared to other pathotypes (Table 5).

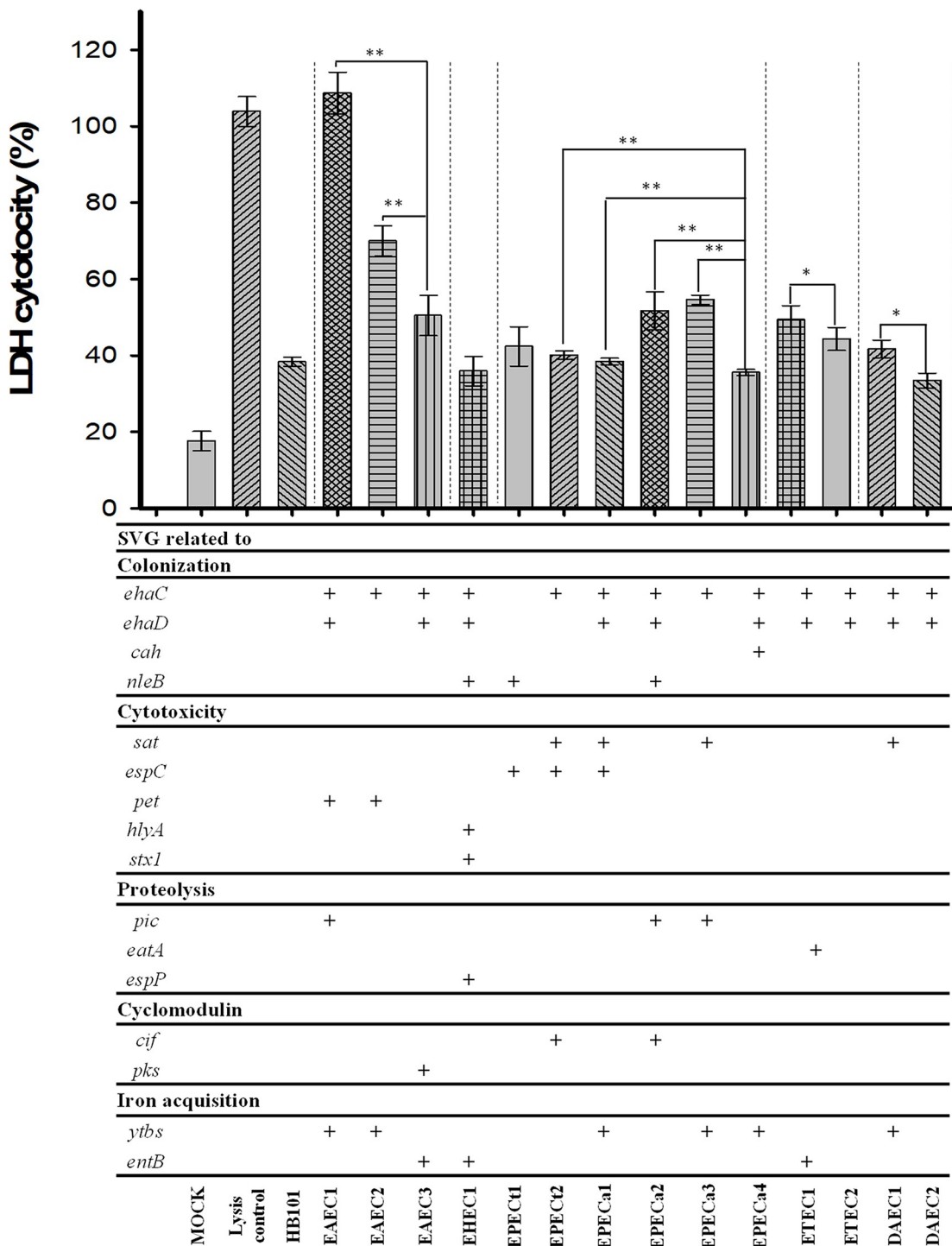

**Fig 2. Cytotoxicity assay of DEC with SVG isolated from children with diarrhea.** HEp-2 cells were infected with several DEC strains. For controls, *E. coli* HB101 was used as a negative control of cytotoxicity, while buffer lysis was used as a positive control. Cytotoxicity was quantified by LDH release. One-way analysis of variance was used to determine the p-value. *: $p < 0.00005$, **: $p < 0.000005$.

**Table 5. Antibiotic resistance of DEC isolated from children with diarrhea.**

| Resistance by antibiotic class | Total DEC | *E. coli* pathotypes | | | | | | | *p*-value |
|---|---|---|---|---|---|---|---|---|---|
| | | aEPEC (%) | EAEC (%) | DAEC (%) | tEPEC (%) | ETEC (%) | EIEC (%) | EHEC (%) | |
| | n = 182 (%) | n = 106 (58.2) | n = 28 (15.3) | n = 14 (7.6) | n = 13 (7.1) | n = 12 (6.5) | n = 8 (4.4) | n = 1 (0.5) | |
| **Penicillin** | | | | | | | | | |
| Ampicillin | 130 (71.4) | 81 (76.4) | 17 (60.7) | 12 (85.7) | 8 (61.5) | 8 (66.6) | 4 (50.0) | 0 (0.0) | 0.07 |
| **β-Lactamases** | | | | | | | | | |
| Amoxicillin-clavulanic acid | 61 (33.5) | 40 (37.7) | 8 (28.5) | 8 (57.1)* | 4 (30.7) | 1 (8.3) | 0 (0.0) | 0 (0.0) | 0.03 |
| Ampicillin sulbactam | 47 (28.5) | 30 (28.3) | 6 (21.4) | 8 (57.1)* | 3 (23.0) | 0 (0.0) | 0 (0.0) | 0 (0.0) | 0.01 |
| **Cephalosporins** | | | | | | | | | |
| Cefazolin | 163 (89.5) | 98 (92.4) | 24 (85.7) | 12 (85.7) | 8 (61.5) | 12 (100)* | 8 (100)* | 1 (100)* | 0.012 |
| Cephatoxime | 67 (36.8) | 33 (31.1) | 17 (60.7)* | 8 (57.1) | 1 (7.6) | 7 (58.3) | 0 (0.0) | 1 (100) | 0.004 |
| Ceftriaxone | 29 (15.9) | 17 (16.0) | 8 (28.5)* | 0 (0.0) | 0 (0.0) | 2 (16.6) | 2 (25.0) | 0 (0.0) | 0.047 |
| Ceftazidime | 17 (9.3) | 13 (12.2) | 3 (10.7) | 0 (0.0) | 0 (0.0) | 1 (8.3) | 0 (0.0) | 0 (0.0) | 0.087 |
| **Monobactams** | | | | | | | | | |
| Aztreonam | 30 (16.4) | 20 (18.8) | 6 (21.4) | 0 (0.0) | 0 (0.0) | 1 (8.3) | 2 (25.0) | 1 (100) | 0.06 |
| **Carbapenems** | | | | | | | | | |
| Imipenem | 14 (7.9) | 4 (3.7) | 4 (14.2) | 0 (0.0) | 3 (23.0) | 3 (25.0)* | 0 (0.0) | 0 (0.0) | 0.019 |
| **Aminoglycosides** | | | | | | | | | |
| Kanamycin | 31 (17.0) | 23 (21.6) | 1 (3.5) | 1 (7.1) | 3 (23.0) | 3 (25.0) | 0 (0.0) | 0 (0.0) | 0.334 |
| Streptomycin | 117 (64.2) | 63 (59.4) | 23 (82.1)* | 11 (78.5) | 7 (53.8) | 7 (58.3) | 6 (75.0) | 0 (0.0) | 0.023 |
| **Tetracyclines** | | | | | | | | | |
| Tetracycline | 89 (48.9) | 54 (50.9) | 9 (32.1) | 11 (78.5)* | 6 (46.1) | 5 (41.6) | 4 (50.0) | 0 (0.0) | 0.019 |
| **Fluoroquinolones** | | | | | | | | | |
| Ciprofloxacin | 26 (14.5) | 17 (16.0) | 6 (21.4) | 0 (0.0) | 0 (0.0) | 3 (25.0) | 0 (0.0) | 0 (0.0) | 0.235 |
| **Quinolones** | | | | | | | | | |
| Nalidixic acid | 72 (39.5) | 47 (44.3) | 11 (39.5) | 6 (42.8) | 1 (7.6) | 3 (25.0) | 3 (37.5) | 1 (100) | 0.119 |
| **Folate inhibitor** | | | | | | | | | |
| Trimethoprim | 101 (55.4) | 65 (61.3) | 15 (53.5) | 11 (78.5) | 4 (30.7) | 2 (16.6) | 4 (50.0) | 0 (0.0) | 0.06 |
| **Phenicols** | | | | | | | | | |
| Chloramphenicol | 13 (7.1) | 11 (10.3) | 0 (0.0) | 0 (0.0) | 0 (0.0) | 0 (0.0) | 2 (25.0)* | 0 (0.0) | 0.044 |
| Resistance by antibiotic number | | | | | | | | | |
| 1 | 6 (3.3) | 4 (3.7) | 1 (3.5) | 0 (0.0) | 1 (7.6) | 0 (0.0) | 0 (0.0) | 0 (0.0) | 0.363 |
| 2 | 18 (9.8) | 6 (5.6) | 3 (10.7) | 1 (7.4) | 3 (23.0) | 2 (16.6) | 3 (37.5)* | 0 (0.0) | 0.033 |
| 3 | 23 (12.6) | 15 (14.1) | 3 (10.7) | 0 (0.0) | 2 (15.3) | 3 (25.0) | 0 (0.0) | 0 (0.0) | 0.18 |
| 4 | 24 (13.2) | 15 (14.1) | 4 (14.8) | 0 (0.0) | 2 (15.3) | 2 (16.6) | 0 (0.0) | 1 (100) | 0.488 |

*(Continued)*

**Table 5.** (Continued)

| Resistance by antibiotic class | Total DEC | *E. coli* pathotypes | | | | | | | p-value |
|---|---|---|---|---|---|---|---|---|---|
| | | aEPEC (%) | EAEC (%) | DAEC (%) | tEPEC (%) | ETEC (%) | EIEC (%) | EHEC (%) | |
| | n = 182 (%) | n = 106 (58.2) | n = 28 (15.3) | n = 14 (7.6) | n = 13 (7.1) | n = 12 (6.5) | n = 8 (4.4) | n = 1 (0.5) | |
| 5 | 20 (11.0) | 14 13.2) | 1 (3.5) | 1 (7.4) | 1 (7.6) | 1 (8.3) | 2 (25.0) | 0 (0.0) | 0.231 |
| 6 | 22 (12.0) | 10 (9.4) | 5 (17.8) | 3 (21.4) | 0 (0.0) | 2 (16.6) | 2 (25.0) | 0 (0.0) | 0.249 |
| 7 | 22 (12.0) | 13 (12.2) | 3 (10.7) | 4 (28.5) | 1 (7.6) | 0 (0.0) | 1 (12.5) | 0 (0.0) | 0.071 |
| 8 | 15 (8.2) | 11 (10.3) | 1 (3.5) | 2 (14.2) | 0 (0.0) | 1 (8.3) | 0 (0.0) | 0 (0.0) | 0.323 |
| 9 | 13 (7.1) | 8 (7.5) | 3 (10.7) | 2 (14.2) | 0 (0.0) | 0 (0.0) | 0 (0.0) | 0 (0.0) | 0.262 |
| 10 | 5 (2.7) | 4 (3.7) | 1 (3.5) | 0 (0.0) | 0 (0.0) | 0 (0.0) | 0 (0.0) | 0 (0.0) | 0.304 |
| 11 | 2 (1.1) | 2 (1.8) | 0 (0.0) | 0 (0.0) | 0 (0.0) | 0 (0.0) | 0 (0.0) | 0 (0.0) | 0.337 |
| 12 | 2 (1.1) | 1 (0.9) | 1 (3.5) | 0 (0.0) | 0 (0.0) | 0 (0.0) | 0 (0.0) | 0 (0.0) | 0.284 |
| 13 | 3 (1.6) | 2 (1.8) | 1 (3.5) | 0 (0.0) | 0 (0.0) | 0 (0.0) | 0 (0.0) | 0 (0.0) | 0.396 |
| 14 | 1 0.5) | 1 (0.9) | 0 (0.0) | 0 (0.0) | 0 (0.0) | 0 (0.0) | 0 (0.0) | 0 (0.0) | 0.582 |
| **Resistance by categories** | | | | | | | | | |
| Resistant to any antibiotic | 145 (79.7) | 105 (99.1) | 27 (93.1) | 13 (92.8) | 0 (0.0) | 0 (0.0) | 0 (0.0) | 0 (0.0) | 0 |
| Sensible | 7 (3.8) | 1 (0.9) | 1 (3.5) | 1 (7.4) | 3 (23.0)* | 1 (8.3) | 0 (0.0) | 0 (0.0) | 0.008 |
| MDR | 80 (43.9) | 44 (41.5) | 12 (42.8) | 3 (21.4) | 8 (61.5) | 9 (75.0)* | 3 (37.5) | 1 (100) | 0.026 |
| XDR | 75 (41.2) | 47 (44.3) | 9 (32.1) | 9 (64.2)* | 7 (53.8) | 1 (8.3) | 2 (25.0) | 0 (0.0) | 0.047 |

MDR = Multidrug-resistant, resistant ≥ 3 different categories of antibiotics; XDR = Extremely drug-resistant ≥ 6 different categories of antibiotics. OR: odds ratio, CI: confidence index

*: statistical significance. Chi-square was used to get statistical significance.

## Discussion

Diarrhea is a disease that causes many deaths in children, especially in developing countries. In this study, 63% of the samples collected from children with diarrhea from different municipalities of Oaxaca were DEC positive, six pathotypes were identified, and combinations of 2 and 3 pathotypes, with aEPEC being the most prevalent; in addition, associations of single infection or pathotype co-infection with age of children and infant symptomatology were found. Furthermore, associations of SVG related to colonization (*nleB*), cytotoxicity (*sat*, and e*spC*), and proteolysis (*pic*) with different pathotypes were found, as well as associations of SVG related to cytotoxicity (*sat*, and *espC*) and proteolysis (*pic*) with *E. coli* pathotypes. Finally, more than 40% of the DEC were MDR and XDR, with tEPEC and DAEC being the most prevalent.

Diarrhea is common in Mexican children; however, many microorganisms can cause diarrhea, including DEC [26]. Therefore, the prevalence of DEC in Mexico may vary by region. Patzi-Vargas et al. (2015) [4] found a high prevalence (56%) of DEC in children with diarrhea in the state of Yucatan, located in southeastern Mexico, data consistent with our findings [4]. In contrast, Angulo-Zamudio et al. (2021) found a prevalence of 18.6% of DEC in children with diarrhea in northwestern Mexico [20]. Similarly, Estrada-Garcia et al. (2009) reported a 16% prevalence of DEC in children <2 years of age in Mexico City [27].

The high DEC prevalence that cause diarrhea in this study could be related by the level of poverty of Oaxaca population; conditions related to poverty as lack education, bad quality of water and low sanitation were associated with infectious diseases, largely with *E. coli*; more-over, the infections in poverty population were related with microorganisms with high antimicrobial resistance [28]. Poverty is a condition present in worldwide and it had related with diarrhea caused by *E. coli* in other countries, for example in United State of America adults in poverty were more likely to suffer STEC infection that non-poverty adults, in case of children they could be suffer DEC infection by transmission person (infected) to children, consumption of contaminated food or contact with animals [29].

Animals or the environment of children develop could had high relevant in prevalence of DEC infection of this study. The interaction of children with environment with feces of pets or cattle could be an infectious focus to get DEC; there are studies in Mexico in which detection of DEC strains were made in those feces [30, 31]. Moreover, Oaxaca is a state with beach, which could be other important focus to get DEC infections. Presence of DEC strains were identified in beach sand of Oaxaca, also in sea animals as turtles and sea lions in others Mexican state; however, this does not rule out that in sea animals of Oaxaca also they are colonized by DEC strains [32–34]. Other important infection focus in which children could gen DEC infection are the rivers of Oaxaca, DEC strains had been identified in rivers of other countries [35]; nonetheless, in Oaxaca has more than 10 rivers and there was not studies related of DEC presence in rivers. Undoubtedly, more studies are needed in Oaxaca state to identify the DEC focus infection to children. Oaxaca is a leading cheese producer in Mexico, in many cases the cheese is made by hand without the necessary safety processes, which has led to cheese quality studies *E. coli* strains had been identified which makes it a source of infection for children of Oaxaca [36].

In this work, aEPEC was the most prevalent isolated pathotype; these data differ from other studies in Mexico [17, 20]. Identify *E. coli* pathotypes is very important, because some pathotypes were observed to be associated with a specific age of children with diarrhea. Similar to this study, Joffre and Iñiguez (2020) found that EAEC was the most common pathotype in 414 children with diarrhea aged 7 to 12 months from Bolivia [37]. Pabst et al. (2003) also associated the presence of EAEC with diarrhea in Swiss children aged 0 to 5 years [38]. In contrast to our findings, Ahmed et al. (2014) found the highest proportion of EAEC in Egyptian children with diarrhea aged 6–11 years compared to all other age groups [39]. Nguyen et al. (2006) identified EPEC as the leading etiologic agent of diarrhea in Australian children aged 16.9 months [40]. Similarly, Chellapandi et al. (2017) found the highest prevalence of typical and atypical EPEC in diarrhea cases of Indian children aged 0–24 months [41]; all these data differ from our study. DEC infection in infant age groups varies from country to country. Some factors may contribute to DEC infection, such as DEC prevalent in the region, children's immune system, nutritional status, and changes in gut microbiota; in addition, children's feeding (mainly <2 years) is entirely dependent on the mother (through breastfeeding), and maternal hygiene may be crucial for mother-to-child transmission of some DEC [42].

DEC infection in children can cause various symptoms, some associated with specific pathotypes. In this study, bloody diarrhea was associated with EHEC. EHEC has many virulence factors that can cause damage to the host; this bacterium colonizes the intestine, where it causes lesions such as destruction of microvilli, intimate attachment of the bacterium to the intestinal cell, and accumulation of polymerized actin under the site of bacterial attachment to form a pedestal-like structure that hollows out individual bacteria, resulting in bloody diarrhea [43, 44]. For the remaining pathotypes, symptoms such as fever, vomiting, persistent diarrhea, mucoid diarrhea, watery diarrhea, and nausea, among others, have been associated with single EPEC, EPEC, or ETEC infection in various studies [41, 45–47]. However, co-infection with

two or three pathotypes may cause more symptoms in children with diarrhea; Khairy et al. (2020) showed that co-infection of EAEC, tEPEC or/and aEPEC presented in higher proportion most of the symptoms presented in children with diarrhea compared to infection by a single DEC. This study agrees with ours as co-infection of different combinations of DEC were the only one associated with diarrhea + other symptoms (1–4) [48].

To cause diarrhea, DEC may need more than their classical virulence factors; in this work, we found associations between some SVG with different pathotypes that could contribute to the development of diarrhea in children in our region. EHEC was associated with *nleB* (related to SVG colonization); others had identified this gene in EHEC [49, 50]. *nleB* is involved not only in the colonization process but also in intestinal cell damage and host mortality; furthermore, this gene has been associated with the ability of EHEC to cause hemolytic uremic syndrome and flare. Apparently, this gene is a linchpin for host initiation, progression, and death, since, in a study testing *nleB* mutant strains in mice, they were unable to cause death compared to wild type [51, 52]. Another SVG (related to cytotoxicity) that we found to be associated with DAEC was *sat*. *sat* is an autotransporter toxin that belongs to the V-type secretion pathway-dependent subfamily of Enterobacteriaceae toxin serine protease autotransporters; this gene has been characterized in *E. coli*; however, it has also been identified in DAECs from children with or without diarrhea, with a higher prevalence in those with diarrhea [53]. sat plays a vital role in the intestinal pathogenesis of DAECs, this gene induces lesions in the binding barrier in Caco-2/TC7 monolayers [54, 55].

The prevalence of *espC* found in diarrhea-producing DAECs is close to 50%, consistent with our results [56, 57]. Another cytotoxicity-related SVG that we found associated with tEPEC was e*spC*. *espC* has been previously described in EPEC, and this gene has multiple functions in EPEC pathogenicity. *espC* in EPEC is an autotransporter not encoded by the locus of enterocyte effacement (LEE); the *espC* protein is released into the interior of cells through the type III secretion system, causing cell rounding and detachment (cytopathic effect); In addition, *espC* has been linked to the ability of EPEC to cause cell death, inducing both apoptosis and necrosis in epithelial cells [58, 59]. The last SVG (proteolysis-related) associated with aEPEC was *pic*. *pic* is a self-transporting serine protease of Enterobacteriaceae typically found in EAEC; in this pathotype, *pic* exhibits proteolytic activity and contributes to mucus hypersecretion to colonize EAEC. However, in aEPEC, Pic has a different activity; for example, Pic agglutinates erythrocytes, cleaves mucin, and degrades complement system molecules. Moreover, in an in vivo model, Pic contributed to the colonization and mucus production of aEPEC in mice [60].

On the other hand, no association was found between cyclomodulin-related SVG and iron acquisition; however, it is noteworthy that DEC that causes diarrhea also harbor cyclomodulins (*pks*, *cif*, and *cnf*), genotoxins that can alter the DNA of intestinal cells and produce tumors [11]. Apparently, SVGs related to colonization, cytotoxicity, and proteolysis play an important role in the DECs that caused diarrhea in Oaxacan children, besides being the only gene cluster associated with diarrhea cases; in the LDH assay, DEC strains with the highest cytotoxic activity were those with one or more SVGs from this gene cluster (colonization, cytotoxicity, and proteolysis).

In addition, DECs isolated as SVGs in this work were associated with *E. coli* phylogroups. The most prevalent phylogroup in DECs was A, although phylogroup A together with B1 are associated with commensal *E. coli*, DEC strains can also colonize children without causing diarrhea as commensal strains; however, changes in the immune system or microbiota in children may contribute to DECs (as commensals) causing diarrhea [20, 61]. In our study, DECs or SVGs (DAEC, *espC*, and *sat*) were associated with phylogroup B2 or D, which are related to strains of high virulence, extraintestinal *E. coli* that cause urinary tract infections, or strains with high antibiotic resistance [62].

Finally, we analyzed the antimicrobial resistance of diarrhea-producing DEC isolated from Oaxacan children, in which we found high levels of antibiotic resistance. The antimicrobial resistance found in this study is similar to DEC isolated from children with diarrhea in northwestern and central Mexico, with approximately or more than 50% of strains resistant to ampicillin, tetracycline, and trimethoprim [17, 63]. In addition, more than 40% of the DECs in this study were MDR or XDR, data consistent with other studies [64, 65]. The high antibiotic resistance presented by DECs in developing countries such as Mexico is related to the indiscriminate use of antibiotics, not only in the clinical setting but also in the food industry, agriculture, and livestock. The high antimicrobial resistance of DECs is alarming, and new strategies should be developed to combat this public health problem [66, 67].

This is the first comprehensive work in Mexico that associates DEC with age, symptoms, SVG, phylogroups (the latter two also with each other), and antibiotic resistance. The limitation is the sample size; with larger samples, we could find more associations between DEC and the variables analyzed in this work.

## Conclusion

This study provides evidence that children with diarrhea in southeastern Mexico were infected with single or co-infected DECs, aEPEC was the most prevalent pathotype; DECs were associated with age and symptoms of the children. In addition, DEC strains (EHEC, DAEC, tEPEC, and aEPEC) were associated with SVG (*nleB*, *sat*, e*spC*, and *pic*), as were E. coli pathotypes with SVG. Finally, increased antibiotic resistance was found in DEC strains. DEC research focused on identifying the clinical and molecular characteristics of this pathogen is necessary; rapid identification of pathotypes, virulence factors, and antibiotic resistance could help to choose the best therapy to eradicate DEC infections in children, reducing the number of mortalities in this vulnerable population.

## Supporting information

**S1 Table. *Escherichia coli* pathotypes and associated virulence genes.**
(DOCX)

**S2 Table. Primers used in this study.**
(DOCX)

**S3 Table. PCR profile of isolate by isolate.**
(DOCX)

## Acknowledgments

The authors thank all the staff of the Oaxaca State Public Health Laboratory, for their technical help. We would like to thank Francisco Martínez-Villa for his assistance with the statistical analysis.

## Author Contributions

**Conceptualization:** Gabriela Tapia-Pastrana, Nidia Leon-Sicairos, Uriel A. Angulo-Zamudio, Adrian Canizalez-Roman.

**Data curation:** Gabriela Tapia-Pastrana, Nidia Leon-Sicairos, Uriel A. Angulo-Zamudio.

**Formal analysis:** Olegario Santiago-Martínez, Lucía C. Gómez-Rodríguez.

**Funding acquisition:** Gabriela Tapia-Pastrana.

**Investigation:** Metztli Rojas-Bautista, Pilar Hernández-Pérez.

**Methodology:** Metztli Rojas-Bautista, Pilar Hernández-Pérez.

**Project administration:** Gabriela Tapia-Pastrana, Uriel A. Angulo-Zamudio, Adrian Canizalez-Roman.

**Software:** Eduardo S. García-López, Nidia Leon-Sicairos.

**Supervision:** Jacobo Montes-Yedra, Alfonso A. Bautista-Avendaño.

**Validation:** Alfonso A. Bautista-Avendaño, Eduardo S. García-López.

**Visualization:** Metztli Rojas-Bautista, Pilar Hernández-Pérez, Eduardo S. García-López.

**Writing – original draft:** Gabriela Tapia-Pastrana, Nidia Leon-Sicairos, Uriel A. Angulo-Zamudio, Adrian Canizalez-Roman.

**Writing – review & editing:** Víctor M. Terrazas-Luna, Jacobo Montes-Yedra.

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
