## [Decision Letter · Decision Letter 0]

29 Jan 2024

PONE-D-23-33103Diarrheagenic Escherichia coli are isolated primarily from children under five years of age with diarrhea in low-income areas of Mexico: Supplementary virulence genes (SVG) and antimicrobial resistance.PLOS ONE

Dear Dr. Tapia-Pastrana,

Thank you for submitting your manuscript to PLOS ONE. After careful consideration, we feel that it has merit but does not fully meet PLOS ONE’s publication criteria as it currently stands. Therefore, we invite you to submit a revised version of the manuscript that addresses the points raised during the review process.

We look forward to receiving your revised manuscript.

Kind regards,

Md Bashir Uddin, PhD

Academic Editor

PLOS ONE

Journal Requirements:

4. Thank you for stating the following financial disclosure:"This work was supported by grants #25501 and #25101 from the Secretaría de Salud through the Hospital Regional de Alta Especialidad de Oaxaca to G.T.P"  

5. We note that your Data Availability Statement is currently as follows: "All relevant data are within the manuscript and its Supporting Information files."

7. We note that Figure 1 in your submission contain [map/satellite] images which may be copyrighted. All PLOS content is published under the Creative Commons Attribution License (CC BY 4.0), which means that the manuscript, images, and Supporting Information files will be freely available online, and any third party is permitted to access, download, copy, distribute, and use these materials in any way, even commercially, with proper attribution. For these reasons, we cannot publish previously copyrighted maps or satellite images created using proprietary data, such as Google software (Google Maps, Street View, and Earth). For more information, see our copyright guidelines: http://journals.plos.org/plosone/s/licenses-and-copyright.

Reviewers' comments:

Reviewer's Responses to Questions

**Comments to the Author**

1. Is the manuscript technically sound, and do the data support the conclusions?

Reviewer #1: Partly

Reviewer #2: Yes

2. Has the statistical analysis been performed appropriately and rigorously? 

Reviewer #1: I Don't Know

Reviewer #2: Yes

3. Have the authors made all data underlying the findings in their manuscript fully available?

Reviewer #1: Yes

Reviewer #2: Yes

4. Is the manuscript presented in an intelligible fashion and written in standard English?

Reviewer #1: No

Reviewer #2: Yes

5. Review Comments to the Author

Reviewer #1: Diarrheagenic Escherichia coli are isolated primarily from children under five years of

age with diarrhea in low-income areas of Mexico: Supplementary virulence genes

(SVG) and antimicrobial resistance.

Gabriela Tapia-Pastrana et al.

Lines 48-52. Can you include any statistics on childhood morbidity/mortality due to diarrhea?

Lines 51-52. List a few of the major causes of childhood diarrhea, globally. Also, please add a reference for this statement.

Line 57. Make sure eaeA+ is italicized.

Line 58. Re-write this sentence to state: In the absence of the EAF plasmid, EPEC are classified as atypical EPEC (aEPEC).

Line 60. Re-write to state: Enteroinvasive E. coli (EIEC) strains encode transcription activators and a type three secretion system (T3SS).

Line 62 and 63. Replace “have” with “encode”.

Paragraphs 2 and 3 need to be made more concise. As they are written now, they sound like a review paper. Please try to find a way to reduce the text. Maybe a table showing the pathotypes and associated genes would help. Im sure that there are many examples of this in the literature.

Lines 81 – 84. Need references for these statements.

Line 91. Put all of the references at the end of the sentence.

Line 94. Remove “our work team” and re-write the sentence to cite the study without mentioning the group.

Line 97. Do not start the sentence with “On the other hand,” this is too casual. Also, many readers will not know where Oaxaca is. How does the poverty level influence DEC prevalence. As statement like this should be reserved for the Discussion and should have a citation.

Lines 109 to 111. Please add a citation for this statement.

Line 140. You mention that “the most representative amplicons were sequenced for control purposes.” How was this done? Also please list the PCR reaction conditions or state where they can be found.

Line 188. Where can we find the breakpoints? How were bacteria determined to be “intermediate resistant”? What diameter was used to identify this?

Line 233. Could you include a supplementary table showing the patient-by-patient or isolate-by-isolate PCR results?

Line 226 What is tEPEC? Please define this earlier in the manuscript.

Figure 2 caption. Please do not include the methods in the caption, at least not to this much detail.

For P-values throughout the text, please use “<” or “>” instead of “:”

Line numbers disappear at the beginning of the Discussion section.

In the sentence “In this study, 63% of children with diarrhea from different municipalities of Oaxaca were DEC positive…” please change to “In this study, 63% of the samples collected from children with diarrhea from different municipalities of Oaxaca were DEC positive …”

You need a reference for this statement: “Diarrhea is common in Mexican children; however, many microorganisms can cause diarrhea.” Also, how does this lead to the next statement, “Therefore, the prevalence of DEC in Mexico may vary by region”? How are these two statements connected?

Do not use First and Middle initials in the in-text references.

The Discussion needs to be re-written so that it is not just a rehash of the results.

There should be some discussion of the potential sources of these infections. Have DEC-like bacteria been found in animals, foods, water in Oaxaca?

The manuscript needs English-language editing, specifically with respect to word-choice.

Reviewer #2: 1-concern the title:

line 1-3, the title is not the representative and redundant title. the title must be self explanatory and it is prefer to be: Virulence genes, antimicrobial resistance profile, phylotyping and pathotyping of Diarrheagenic Escherichia coli isolated from children in southwest Mexico

2-Concern the abstract :

line 28, etiological factors: it is best to etiology

line 29, word contain: it is best to be, have

line 37, concern the of aEPEC, EAEC, DAEC, tEPEC, ETEC, EIEC, or EHEC,: pls. mention the percentage for each and also for co-infection.

line 39-40, authors mentioned that, SVG related to colonization (nleB), cytotoxicity (sat and espC),

and proteolysis (pic) were associated with DEC strains: who can you approve the association?

line 40-41, authors sated that, whereas espC, sat, and pic were associated with E. coli pathotypes: it is not clear and confused

line 42, the authors mentioned some strains as XDR, XDR strains mean that, it is resist at least one antibiotic from each class for at least all classes (mentioned in CLSI) except 2 but unfortunately the authors not used all classes of antibiotics as mentioned in CLSI so cannot regard its as XDR

finally the authors study the phylogroups of E. coli but in abstract nothing were mentioned concern the phylogroups of DEC

6. PLOS authors have the option to publish the peer review history of their article (what does this mean?). If published, this will include your full peer review and any attached files.

Reviewer #1: No

Reviewer #2: **Yes: **Prof. Dr. Hussein O.M. Al-Dahmoshi

---

## [Author Response · Author response to Decision Letter 0]

19 Feb 2024

Response to Reviewers

Reviewer #1: Diarrheagenic Escherichia coli are isolated primarily from children under five years of age with diarrhea in low-income areas of Mexico: Supplementary virulence genes

(SVG) and antimicrobial resistance.

Gabriela Tapia-Pastrana et al.

Response: Thank you for taking the time to review the new version of the manuscript, and we will address all your comments below.

Lines 48-52. Can you include any statistics on childhood morbidity/mortality due to diarrhea?

Response: Thank you for your recommendation. We agree, and the child mortality data for diarrhea has been added as follows: More than 446,000 child deaths from diarrhea have been reported worldwide (line 49).

Lines 51-52. List a few of the major causes of childhood diarrhea, globally. Also, please add a reference for this statement.

Response: Thank you for your comment, and the major causes of diarrhea in children have been added (lines 52-53).

Line 57. Make sure eaeA+ is italicized.

Response: This sentence was deleted due to the recommendation to add a supplementary Table 1 listing E. coli pathotypes and their associated genes.

Line 58. Re-write this sentence to state: In the absence of the EAF plasmid, EPEC are classified as atypical EPEC (aEPEC).

Response: This sentence was deleted due to the recommendation to add supplementary Table 1, which lists the pathotypes of E. coli and their associated genes.

Line 60. Re-write to state: Enteroinvasive E. coli (EIEC) strains encode transcription activators and a type three secretion system (T3SS).

Response: This sentence was deleted due to the recommendation to add supplementary Table 1, which lists the pathotypes of E. coli and their associated genes.

Line 62 and 63. Replace “have” with “encode”.

Response: This sentence was deleted due to the recommendation to add Supplemental Table 1, which lists the pathotypes of E. coli and their associated genes.

Paragraphs 2 and 3 need to be made more concise. As they are written now, they sound like a review paper. Please try to find a way to reduce the text. Maybe a table showing the pathotypes and associated genes would help. Im sure that there are many examples of this in the literature.

Response: This is a good point, we agree, and to improve the introductory section, the paragraphs have been modified, and Supplementary Table 1 has been added with virulence factor genes that classify DEC into six different pathotypes (lines 56-58).

Lines 81 – 84. Need references for these statements.

Response: We apologize for the error; references to the statement have been added on lines 70-73.

Line 91. Put all of the references at the end of the sentence.

Response: Thank you for your comment. We agree, and the references have been moved to the end of the sentence (line 80).

Line 94. Remove “our work team” and re-write the sentence to cite the study without mentioning the group.

Response: Thank you for your recommendation. To improve the Introduction section, we have removed "our teamwork" and added Angulo-Zamudio et al. (2021) (line 83).

Line 97. Do not start the sentence with “On the other hand,” this is too casual. Also, many readers will not know where Oaxaca is.

Response: Thank you for your comment. to improve the introduction, "on the other hand" has been changed to "furthermore" (line 86).

How does the poverty level influence DEC prevalence. As statement like this should be reserved for the Discussion and should have a citation.

Response: Thank you for your comment, we agree, and the discussion related to poverty and DEC prevalence has been added (lines 336-374).

Lines 109 to 111. Please add a citation for this statement.

Response: Thank you for your recommendation, and the reference has been added as you kindly suggested (line 99).

Line 140. You mention that “the most representative amplicons were sequenced for control purposes.” How was this done? 

Response: Thank you for your observation. we have selected the most representative amplicons of the genes that determine each of the DEC, Cyclomodulin, or SVG strains in the PCR reactions to sequence them to ensure the positive result of each of the DEC, Cyclomodulin, or SVG strains.

Also please list the PCR reaction conditions or state where they can be found.

Response: Thank you for this comment; the melting temperature data for all primers used in the PCR reactions are listed in Supplementary Table 2.

Line 188. Where can we find the breakpoints? How were bacteria determined to be “intermediate resistant”? What diameter was used to identify this?

Response: Thank you for your comment. we followed the Clinical Laboratory Standard Institute guidelines for determining DEC strains as resistant, intermediate, or sensitive (as we show in lines 201-202); however, for visual and important reasons, we decided to show only the resistance of the bacteria in the table.

Line 233. Could you include a supplementary table showing the patient-by-patient or isolate-by-isolate PCR results?

Response: Thank you for your recommendation, we agree and to improve this section, Supplementary Table 3 has been added with the isolate-by-isolate PCR results as follows: '' '' '' '' In addition, the isolate-by-isolate PCR profile is shown in Supplementary Table 3'' (lines 220-221).

Line 226 What is tEPEC? Please define this earlier in the manuscript.

Response: Thank you for your comment, and in the Materials and Methods sections ''Molecular identification of DEC strains'' we have added the sentence ''detected both typical (tEPEC, eae+ and bfp+) and atypical (aEPEC, eae+ and bfp-) EPEC strains'' (line 137).

Figure 2 caption. Please do not include the methods in the caption, at least not to this much detail.

Response: Thank you for your recommendation, and to improve Figure 2, the method was summarized (lines 302-306). 

For P-values throughout the text, please use “<” or “>” instead of “:”

Response: Thank you for your comment. we did use "<" or ">" when the p-value was lower or higher than a certain value; however, most of the values reported in the manuscript were the exact value of the result of the statistical analysis.

Line numbers disappear at the beginning of the Discussion section.

Response: We apologize for this error. To improve the discussion section, we have added the line number.

In the sentence “In this study, 63% of children with diarrhea from different municipalities of Oaxaca were DEC positive…” please change to “In this study, 63% of the samples collected from children with diarrhea from different municipalities of Oaxaca were DEC positive …”

Response: Thank you for your recommendation, we agree, and we changed the sentence as you kindly suggested (lines 351-352).

You need a reference for this statement: “Diarrhea is common in Mexican children; however, many microorganisms can cause diarrhea.”

Response: Thank you for your comment, and to improve the discussion section, we have added the reference to the sentence ''Diarrhea is common in Mexican children; however, many microorganisms can cause diarrhea'' (lines 359-360).

Also, how does this lead to the next statement, “Therefore, the prevalence of DEC in Mexico may vary by region”? How are these two statements connected?

Response: Thank you for your comment. We agree, and to improve this paragraph, we will modify the sentence as follows: '' '' Diarrhea is common in Mexican children; however, many microorganisms can cause diarrhea, including DEC'' (lines 359-360).

Do not use First and Middle initials in the in-text references.

Response: Thank you for your comment, and the middle initials have been removed from the discussion section as you kindly suggested.

The Discussion needs to be re-written so that it is not just a rehash of the results.

Response: Thank you for your suggestion, and to improve the discussion section, we have added topics related to poverty and DEC infection (lines 366-374), as well as potential sources of DEC infection in Oaxacan children (lines 375-388).

There should be some discussion of the potential sources of these infections. Have DEC-like bacteria been found in animals, foods, water in Oaxaca?

Response: Thank you for your recommendation; we agree, and to improve the discussion section, we have added the potential sources of DEC infection in Oaxacan children (lines 375-388).

The manuscript needs English-language editing, specifically with respect to word-choice.

Response: Thanks for your recommendation. The entire manuscript has been grammatically corrected in English.

 

Reviewer #2: 

Response: Thank you for taking the time to review the new version of the manuscript, and we will address all of your comments below.

1-concern the title:

line 1-3, the title is not the representative and redundant title. the title must be self explanatory and it is prefer to be: Virulence genes, antimicrobial resistance profile, phylotyping and pathotyping of Diarrheagenic Escherichia coli isolated from children in southwest Mexico

Response: Thank you for your comment; we agree and have changed the title of the manuscript as you kindly suggested.

2-Concern the abstract :

line 28, etiological factors: it is best to etiology

Response: Thank you for your comment, and "etiological" has been changed to "etiology" as you kindly suggested (line 27).

line 29, word contain: it is best to be, have

Response: Thank you for your comment, we agree and have changed the word as you suggested.

line 37, concern the of aEPEC, EAEC, DAEC, tEPEC, ETEC, EIEC, or EHEC,: pls. mention the percentage for each and also for co-infection.

Response: Thank you for your recommendation. To improve the summary section, we have added the percentage of DEC and coinfection as you kindly suggested (lines 35-37).

line 39-40, authors mentioned that, SVG related to colonization (nleB), cytotoxicity (sat and espC),

and proteolysis (pic) were associated with DEC strains: who can you approve the association?

Response: Thank you for your comment, and to improve the abstract section, we have modified the sentence as follows: ''(nleB-EHEC), cytotoxicity (sat-DAEC and espC-tEPEC), and proteolysis (pic-aEPEC) were associated with DECs strains'' (lines 39-40). In addition, associations were performed by chi-square test.

line 40-41, authors sated that, whereas espC, sat, and pic were associated with E. coli pathotypes: it is not clear and confused

Response: Sorry for the error. To clarify this section, we have removed this sentence, and it reads as follows: ''(nleB-EHEC), cytotoxicity (sat-DAEC and espC-tEPEC) and proteolysis (pic-aEPEC) were associated with DECs strains'' (lines 39-40).

line 42, the authors mentioned some strains as XDR, XDR strains mean that, it is resist at least one antibiotic from each class for at least all classes (mentioned in CLSI) except 2 but unfortunately the authors not used all classes of antibiotics as mentioned in CLSI so cannot regard its as XDR

Response: Thank you for your comment, this is a good point, and CLSI standards were used to determine whether or not DEC strains were resistant; the classification of Magiorakos et al. 2012, where they indicate that isolates resistant to ≥ 3 different categories of antibiotics were classified as multidrug-resistant (MDR), and those resistant to ≥ 6 different categories of antibiotics were classified as extremely drug-resistant (XDR).

finally the authors study the phylogroups of E. coli but in abstract nothing were mentioned concern the phylogroups of DEC

Response: Thank you for your comment, and to improve the summary section, we have added information on pathotypes as follows: E. coli phylogroup A was the most frequent, and some pathotypes (aEPEC - A, DAEC - B) and SVG (espC - B2, and sat - D) were associated with the phylogroups (lines 40-41).

---

## [Editor Report · Decision Letter 1]

26 Feb 2024

Virulence genes, antimicrobial resistance profile, phylotyping and pathotyping of Diarrheagenic Escherichia coli isolated from children in Southwest Mexico.

PONE-D-23-33103R1

Dear Dr. Gabriela,

We’re pleased to inform you that your manuscript has been judged scientifically suitable for publication and will be formally accepted for publication once it meets all outstanding technical requirements.

Kind regards,

Md Bashir Uddin, PhD

Academic Editor

PLOS ONE
---

## [Editor Report · Acceptance letter]

1 Mar 2024

PONE-D-23-33103R1 

PLOS ONE

Dear Dr. Tapia-Pastrana, 

I'm pleased to inform you that your manuscript has been deemed suitable for publication in PLOS ONE. Congratulations! Your manuscript is now being handed over to our production team.

Kind regards, 

on behalf of

Dr. Md Bashir Uddin 

Academic Editor

PLOS ONE